# The Data Sensor Hub (DaSH): A Physical Computing System to Support Middle School Inquiry Science Instruction

**DOI:** 10.3390/s21186243

**Published:** 2021-09-17

**Authors:** Alexandra Gendreau Chakarov, Quentin Biddy, Colin Hennessy Elliott, Mimi Recker

**Affiliations:** 1Computer Science and Science Education, San José State University, San Jose, CA 95125, USA; alexandra.chakarov@sjsu.edu; 2Institute of Cognitive Science, University of Colorado, Boulder, CO 80309, USA; 3Instructional Technology & Learning Sciences, Utah State University, Logan, UT 84322, USA; mimi.recker@usu.edu

**Keywords:** sensor use in education, programmable sensors in education, inquiry science education

## Abstract

This article describes a sensor-based physical computing system, called the *Data Sensor Hub* (*DaSH*), which enables students to process, analyze, and display data streams collected using a variety of sensors. The system is built around the portable and affordable BBC micro:bit microcontroller (expanded with the gator:bit), which students program using a visual, cloud-based programming environment intended for novices. Students connect a variety of sensors (measuring temperature, humidity, carbon dioxide, sound, acceleration, magnetism, etc.) and write programs to analyze and visualize the collected sensor data streams. The article also describes two instructional units intended for middle grade science classes that use this sensor-based system. These inquiry-oriented units engage students in designing the system to collect data from the world around them to investigate scientific phenomena of interest. The units are designed to help students develop the ability to meaningfully integrate computing as they engage in place-based learning activities while using tools that more closely approximate the practices of contemporary scientists as well as other STEM workers. Finally, the article articulates how the DaSH and units have elicited different kinds of teacher practices using student drawn modeling activities, facilitating debugging practices, and developing place-based science practices.

## 1. Introduction

Advanced computing technologies have become integral to all aspects of science, technology, and engineering research and practice [1]. One particular form of technology, sensors, is playing an increasing role in automatically detecting and recording a wide range of environmental properties. When the resulting streams of data are further processed by computers to power information displays or actuators, they are often referred to as a kind of physical computing.

Just as sensors and physical computing have revolutionized science and engineering, so too do they offer the potential for students to engage in meaningful scientific inquiry activities in ways that resemble the work of contemporary scientists [2,3,4]. With growing emphasis on providing computer science education to all students [5], these technologies also enable instructional activities that integrate computing in deep and meaningful ways. This is especially true as sensors and microcontrollers drop in price with the rise in home technology hobbyist and maker movements [6]. Further broadening their appeal, many microcontrollers can be programmed with simple visual block-based programming languages. These drag-and-drop programming environments provide a “low floor” entry point for students, meaning novice students can more easily participate in computing and programming activities [7].

Educational physical computing systems have the potential to expand the kinds of computing concepts students can engage in at young ages. Recent work has examined how sensors and physical computing can be integrated in instructional environments [8,9]. For example, DesPortes and DiSalvo [10] studied how novices learned to program the Arduino using simple breadboarding and programming tools. Wagh et al. [11] documented how students learn key computational practices in physical computing, specifically learning to use the hardware, software, and to debug problems across the two. Similarly, Kafai et al. [12] studied how children learned to build and debug electronic textiles (or e-textiles) using the LilyPad Arduino and crafting materials such as conductive thread.

Finally, Hardy et al. [13] analyze how a high school student used sensor-based tools to investigate personally meaningful science questions. They describe how the learning experiences foregrounded the use of sensors to collect data, which can highlight for students how simply collecting data involves a series of consequential decisions. In this way, it can help students learn that data collection is not a neutral act and, instead, is driven by the core needs and intentions of the investigator.

Building on this line of research, the purpose of this article is to describe a sensor-based physical computing system, called the Data Sensor Hub (DaSH), which integrates low-cost, portable, and easily programmable technologies to support students in engaging in a range of science inquiry activities.

Unlike previous work, we also designed several instructional units intended for middle school science classes that integrate use of the DaSH. These units are aligned with core science standards in the U.S., and thus, are meant to be used by a wide range of students. We summarize findings from over four years of classroom use of the DaSH and accompanying instructional units by over 30 teachers and their 3000 students in the Western United States [14,15,16]. We present several themes emerging from this work, focusing on how teachers supported students in engaging in place-based inquiry learning activities while learning to use a powerful data-driven tool for scientific inquiry in ways that more closely approximate the practices of modern scientists.

## 2. Materials and Methods

This section details the iterative development process of the DaSH over three design cycles. The final version of the DaSH is a physical computing system that uses the BBC micro:bit to enable students to collect, analyze, and display data streams using a variety of sensors developed in conjunction with our partners at SparkFun Electronics.

### 2.1. The Data Senor Hub (DaSH)

Several design considerations informed decisions in selecting the components and designing the DaSH for use by middle school students. These were:Cost: As a technology for public schools, it must be affordable.As students will collect environmental data from many locations, the system must be portable and easily configured. Further, as students will manipulate the technology, its components must be robust.Easily programmable by using a simple visual, drag-and-drop, block-based language. Similarly, as students gain proficiency, students have more advanced programming options available.A variety of sensors: To support students in a range of engaging activities, a wide range of sensors must be available.A variety of output options: Simply collecting sensor data is not sufficient. Students need means to control and visualize processed sensor data streams.

These design considerations are complex and, as with any new technology, design modifications grounded in real-world use improve its design. To this end, this section describes three versions of the DaSH, each developed as part of an iterative design cycle. After each version was tested in classrooms, researchers and teachers examined how the current version met the design criteria and recommended changes to increase alignment with the criteria. The largest change came between the first version and second versions of the DaSH when the microcontroller controlling the system was changed.

#### 2.1.1. DaSH Version 1

Researchers, teachers and school administrators partnered with SparkFun Electronics to create a physical computing system that met the design criteria described above. For the first iteration, the design team knew it would be challenging to satisfy all the criteria, so we focused on getting a device that could collect large amounts of data from the local environment into classrooms during the first year of the project. Initially, we worked to create a custom sensor system built using an ESP32 microcontroller and i2C sensors that could be attached using 4-pin JST connectors. The microcontroller was preprogrammed by the SparkFun engineers based on feedback from the teacher partners. While this system allowed students to easily collect large amounts of environmental data from locations all over their school [17] demonstrating robustness, it was quite expensive (~EUR 145 per kit) and it proved challenging to satisfy the additional criteria without greatly increasing the cost of an already expensive kit.

#### 2.1.2. DaSH Version 2

Working with SparkFun Electronics, the research team and school administrators explored alternative, commercially available technology that we could modify and extend for our purposes. Designed by the BBC for use in education, we selected the micro:bit as the base microcontroller. The micro:bit has been in production since 2016, with over 5 million distributed at a cost of approximately EUR 12 per micro:bit. The micro:bit is pocket sized, thus easily transportable. For power, it requires either: (1) two AA batteries, (2) a USB connection to a computer, or (3) a power adapter connected to a wall outlet.

In terms of input, the micro:bit has two input buttons and several on-board sensors which measure light and temperature as well as a magnetometer and accelerometer. For output, it has a 5 × 5 LED light display where each LED is programmable. To expand the range of functionalities, the micro:bit can communicate with other micro:bits using radio waves to send and receive messages and data from other micro:bits. Thus, students can create a group of micro:bits that communicate information to perform different tasks (e.g., one or more micro:bits serve as data collectors that send the information to a central micro:bit for processing and display). The ability to create a network of micro:bits allows students to explore and program devices on the Internet of Things.

MakeCode [18] serves as the programming environment for controlling the micro:bit. The open source environment is supported by Microsoft as a browser-based programming environment which enables users (students) to write programs in block and text-based languages that can be downloaded via USB onto the micro:bit’s flash memory (or via Bluetooth and the app if using a smartphone or tablet). The MakeCode environment also provides a simulation environment where students can test their code. The ability to use MakeCode in a browser removes the hurdle of installing additional software on school computers, which allows for ease of use on cloud-based computers (e.g., Chromebooks) and offers less issues when scaling up educational interventions. Considering our target audience was middle school students, the DaSH and instructional units described in Section 3 were designed primarily to rely on block-based programming, as it provides a simple and visual entry point for students [7]. As students become more familiar with programming concepts, they can progress to the more advanced features included in MakeCode such as the creation of arrays, functions, and even importing external extensions. In addition, they can begin the transition to the more powerful text-based programming languages, JavaScript and Python, thereby taking advantage of the “high ceiling” and “wide walls” [19] in this programming environment.

While the micro:bit contains several onboard sensors to support data collection and LEDs to support data visualization, more sensors and actuators need to be added in accordance with the design guidelines to support a broad range of scientific investigations. Connecting a breadboard to the micro:bit could support these goals. However, implementing breadboarding activities in middle school science classes is challenging and time consuming for teachers.

#### 2.1.3. DaSH Final Version

To create a more user-friendly expansion of the micro:bit targeted towards data collection and display, researchers and teachers continued their collaboration with SparkFun Electronics to create the gator:bit. The gator:bit makes additional pins on the micro:bit accessible for alligator clips (see Figure 1). The use of alligator clips obviates the need for soldering or breadboarding. The gator:bit also has a speaker and 5 programmable neopixel LEDs that students can use to create displays with lights and sound.

To determine an initial set of additional sensors to add, researchers and teachers searched through available sensors and brainstormed questions about scientific phenomena that these sensors could help answer. Engineers at SparkFun Electronics then created five alligator clippable sensors, including (1) an environmental sensor that measures temperature, humidity, barometric pressure, carbon dioxide, and total volatile organic compounds, (2) a soil moisture sensor, (3) a sound sensor, (4) a UV sensor, and (5) a particle sensor (see Figure 1). All of the sensors except the soil moisture sensor are I2C sensors and can be daisy-chained together, which enables multiple sensors to be used at one time. Each of the sensors has their own Makecode blocks so students can control the data collection process. To support the long-term collection of data, a real time clock and data logger that can save data to an SD card can also be attached to the gator:bit using alligator clips and controlled through MakeCode programs.

The DaSH can be purchased as a kit for approximately EUR 90 per kit. This is not as low cost as would be ideal, but the micro:bit and gator:bit cost EUR 28 and additional sensors can be purchased a la carte with cost ranging between EUR 4 and EUR 20, decreasing the initial cost and enabling teachers to slowly build up kits. In addition, many existing kits that provide similar functionality cost well over EUR 100 per kit (see Table 1).

### 2.2. Processing and Displaying Sensor Data Streams

A key aspect of collecting sensor data streams is to address personally meaningful questions for students by processing the data to communicate, display, or visualize relevant information. Furthermore, sensor data can be processed to trigger an action in the world using the built-in speaker and a neopixel array to produce simple data displays. In this way, sensor technologies can help make the invisible visible in what can be particularly powerful learning moments for students.

Figure 2 shows three examples, created by participating teachers, where data are collected from the sensors, processed, and displayed using the micro:bit and gator:bit display and a neopixel LED strip. In the image on the left (Figure 2a), the micro:bit is programmed to measure the moisture in the soil of a plant. The strip of 60 LEDs monitors the moisture with more lights indicating a higher percentage of moisture in the soil. When the moisture falls below a certain percentage, the speaker on the gator:bit plays a tone.

The center image (Figure 2b) depicts a micro:bit that is programmed to control one neopixel for each sensor value (sound, temperature, humidity, barometric pressure, carbon dioxide, and total volatile organic compounds). The neopixel is green when the value is within the acceptable range, the neopixel is blue if the value is below the acceptable range, and the value is red if it is above the acceptable range. The teacher who created this chose the acceptable range values based on research and her personal preference. When button A on the micro:bit is pressed, the exact temperature scrolls across the micro:bit LEDs and when button B is pressed, the exact humidity scrolls across the micro:bit LEDs.

In the image on the right (Figure 2c), the micro:bit was programmed to communicate information about four environmental conditions (temperature, humidity, carbon dioxide level, and sound level) using a strip of 30 LED lights and the 5 by 5 LED light array on the micro:bit. The LED lights on the micro:bit display the variable being measured by showing a letter (e.g., T for temperature) followed by the actual value recorded by the sensor. Each environmental condition is programmed to have its own color associated with it (e.g., blue for humidity) and the strip of 30 LEDs lights up so that as the value of the environmental condition increases, more LED lights are lit to indicate this increase. Additionally, if a predetermined threshold is reached, such as when the noise in the classroom reaches a certain level, the LED light strip blinks.

These displays (Figure 2) represent just a few ways that the DaSH can be used to collect, process, and display data streams. Students can explore this functionality to tell stories with their data and use it to create visual representations of scientific phenomena.

## 3. Instructional Applications and Units

This section details two instructional units that integrate the DaSH into middle school curricula: the Sensor Immersion Unit and the Ecosystems Composting Unit. Participating middle school teachers were directly involved in co-designing and implementing these units in their classrooms. Both units were designed for science classrooms, although they have been adapted for use in STEM classrooms. These units are publicly available on our website (https://www.colorado.edu/program/schoolwide-labs/, accessed on 15 September 2021).

### 3.1. Instructional Unit Design Strategy

We designed several instructional units, in collaboration with middle school teachers, that integrate the DaSH [14]. These units are inquiry-oriented, an increasingly common approach in science instruction where students ask their own questions and solve personally relevant problems [20]. In our context, this means that teachers facilitate students asking questions about a scientific phenomenon. Units consist of connected activities to gather evidence—in our case, using sensor technologies—to address students’ questions.

An important first step in developing inquiry-oriented instructional units for science education is selecting a phenomenon to anchor the unit. A scientific phenomenon is something that can be observed in real life, relies on scientific knowledge to learn about, and requires constructing explanations using observations and marshalling evidence [21,22]. Some examples of phenomena include how the moon affects tides, what happens during a car crash, or how a maglev train works.

After selecting a compelling phenomenon, the teachers and researchers then work together to predict students’ questions about the phenomenon and provide a sequence of activities that sustain coherence and interest throughout the unit [23,24]. Using a question prioritization process, individual lessons in the unit are developed that correspond to these questions. In particular, these lessons guide students in collecting evidence to address their questions, draw conclusions based on the evidence, and construct scientific arguments that explain particular phenomena [2,25].

In this section, we describe two such units. The units follow the same set of instructional routines in order to support students in investigating phenomena by having them first ask questions about it and then design and program their DaSH to collect sensor data (evidence) to address their questions and draw conclusions from evidence. In this way, students are supported in using the DaSH to ask and address meaningful questions about the world around them.

### 3.2. Sensor Immersion Unit

Figure 3 depicts the framing of the Sensor Immersion Unit as designed with participating teachers. In this unit, students engage with these questions, activities, and explorations over the course of five lessons (which could last more than one class period). Below, we describe the method of each designed lesson.

#### 3.2.1. Lesson 1 and 2

In an ideal setup, DaSH data displays, such as what is shown in Figure 2, have been running since the start of the school year in teachers’ classrooms. This drives students’ curiosity as they interact with the technology and helps them consider the question for the unit: “how can sensors help us understand and communicate information about the world around us?” The teacher anchors the unit with a discussion about what students notice and wonder about the technology. As a key instructional routine, students are asked to hand draw an initial explanatory model of the DaSH that they will revisit and revise at the end of each lesson as they figure out more about how the sensor system works. These initial models serve to prompt additional student questions as they try to make sense of their own DaSHs in future lessons.

After they create their hand drawn models, the students develop a set of questions about the DaSH displayed in their classroom, collaboratively categorize those questions, and determine through a discussion which group of questions to investigate and answer first. We have found that student questions generally fall into three categories: (1) what are the different parts of the system, (2) how do the different parts of the system communicate with each other, and (3) how does the system know what to do? Students most frequently decide to investigate the parts of the system first as this is the easiest entry point into understanding the DaSH.

Students then begin to investigate the different parts of the DaSH, focusing on the micro:bit, gator:bit and sensors. Students engage in an introductory programming experience by creating a simple program to measure the strength of magnetic fields and display the value on the micro:bit LED array screen. Students are then able to use their newly programmed micro:bit magnetometer to measure the strength of magnetic fields of different objects in the classroom (e.g., computers, tables, books, etc.). After the students have completed the magnetometer activity, each group chooses one of the available sensors to explore (sound sensor, environmental sensor (temperature, CO2, VOCs, humidity, and/or pressure), soil moisture sensor, UV sensor, particle sensor). They then use provided programming tutorials for their selected sensor, which include scaffolds to support students to figure out how to program their sensor system. Students are then able to collect and display their data numerically on the micro:bit.

#### 3.2.2. Lesson 3 and 4

After students have learned how to collect data using the sensors during the previous lessons, they then want to figure out how to connect these data to the display lights and/or sound as actuators. This process introduces students to programming conditional logic and has the potential for the introduction of variables and loops. Students use provided programming tutorials that scaffold them to go through the process of building a simple data display. During this lesson, students are able to customize their DaSH display based on how they want to communicate and visualize the information in the data streams.

Students then come together to share their findings about their specific sensor with the entire class, including the different types of data each sensor can measure. They also show and explain their MakeCode programs, what their programs are supposed to do, what their program looks like, how the system is wired and challenges they encountered. The goal is for students to recognize similar patterns in the different programs and in the assembly processes in order to realize the commonalities across the sensors, such as how they are wired and patterns in the block coding. Students come to understand that once they know how to use one sensor, they can easily do some work to be able to use any sensor to answer their questions. This allows the students to apply their understanding of the sensor system and related programming to design new systems to answer new questions and solve new relevant scientific problems.

Students conclude this lesson by working together to draw both a final individual model of their system and a whole class consensus explanatory model of the classroom data display.

#### 3.2.3. Lesson 5

Students work collaboratively to brainstorm other questions that the DaSH might help them answer. They then complete a transfer assessment task where they are introduced to a new phenomenon and are asked to describe how the DaSH could help them answer some of the questions they have about the new phenomenon. As a part of this transfer assessment task, they also work to reframe questions so that they can be investigated using the sensor system. At this point, teachers are able to introduce other units where students are able to now see the DaSH as a scientific tool to figure out new phenomena and/or solve new problems.

### 3.3. Compost Ecosystem Unit

In the Composting Ecosystems Unit (Figure 4), students investigate matter and energy cycling through a vermicompost system, using the DaSH to monitor the system. Students investigate questions they have about composting and use sensors to monitor environmental conditions inside worm compost bins with different conditions in order to determine which conditions produce compost that can be used for growing food. Students develop explanatory models around how matter and energy are changing, flowing and cycling through the compost system as food scraps are converted into compost. Students also develop ideas about: how to measure and analyze environmental conditions in a closed system using the DaSH and how to communicate information using evidence about vermicomposting to propose a composting program.

#### 3.3.1. Lesson 1 and 2

Students discuss how urban gardening is a way to improve urban ecosystems, diversity, and can help beautify neighborhoods. Students observe information about urban gardening and the difficulties associated with growing plants in urban settings, especially in soils containing lots of clay. Finally, students reason with what is needed to grow food in an urban setting, specifically what has to be done to improve the soil.

Next, students observe worm composting bins and investigate information on how composting can produce material that can be added to soil to make it better for growing plants. Students construct initial models to show how they think composting with worms (vermicomposting) works to change food scraps into “good” nutrient rich soil. The students begin to ask questions that they can investigate using the DaSH about how composting works and what factors make a worm compost system work to produce the best compost in the shortest amount of time.

#### 3.3.2. Lesson 3 and 4

Through a whole group consensus building discussion, students discuss their ideas about how a compost bin works and brainstorm possible plans for testing the efficiency and effectiveness of vermicompost bins. The class begins by using a common soil test kit to measure the “quality” of the compost by looking at such characteristics as pH, Phosphorus, and Nitrogen content. Students plan investigations with the DaSH to determine what conditions in a compost system result in “good” compost that can be used to grow food and what data they should collect using the sensors (see Figure 5) (i.e., temperature, humidity, CO2, levels, Volatile Organic Compound levels and soil moisture), and how they plan to collect the data with the DaSH. Students then prepare and conduct their investigations.

#### 3.3.3. Lesson 5 and 6

After the DaSH has had sufficient time to collect data about the vermicompost bins for a few days, the students download their datasets and view them via spreadsheets. Students organize and analyze the data they have collected to begin to make sense of what happened in their compost system. Students test the soil in each vermicompost bin again, looking at pH, Phosphorus and Nitrogen content to determine which bin might have the “best” compost that would help plants grow. Students use their collected and organized DaSH data to create visualizations such as tables, graphs, and other types of visualizations to better make sense of their data and so that they can communicate their findings to others.

Students construct a proposal for a schoolwide composting program to share with the school administration. Students research other school composting programs to see what works best. Students use their DaSH data and findings from the investigations along with their research to explain what composting is, how it works, how a composting program could be implemented at their school, the reasons for creating such a composting program in their specific school and neighborhood, and how the DaSH could be used to monitor the composting program.

## 4. Classroom Experience Findings

Over the course of four years of classroom implementations, we examined how over 30 teachers used the DaSH and the instructional units with over 3000 students. In our investigations, we collected a range of data sources, including teacher surveys, video recordings of classroom experiences, student and teacher interviews, and artifacts created by students. Several themes that emerged across analyses of these data include:How students come to understand and think about sensors as a powerful data-driven tool for scientific inquiry through an analysis of their hand-drawn models;How the instructional context naturally led to problems and bugs as students tinkered with the DaSH and teachers’ approaches supported students in solving them;How the DaSH and instructional units supported meaningful place-based inquiry for students.

### 4.1. Exploring How Students Think about the DaSH through Hand-Drawn Models

A key component of the Sensor Immersion Unit involves students creating hand-drawn models of the DaSH. Prior research illustrated how these models can help teachers learn how their students are thinking about the topic [26]. Including this modeling component in the Sensor Immersion Unit provides an opportunity for teachers and researchers to see the big ideas that students are taking away from their initial work with the DaSH.

We completed an analysis of 213 student models from 4 middle school science classrooms [14]. The analysis began with our teacher partners examining a subset of the models to identify themes in students’ thinking, designed after work carried out in [14]. The subset represented the range of student-created models: from extreme detail to something akin to a drawing. The teachers identified four main ideas represented in the models: (1) the components of the DaSH, (2) the flow of information within the DaSH and between the environment and the DaSH, (3) descriptions of how the information was displayed and (4) description of how programming affects the DaSH. The researchers then created and validated a coding scheme to examine the depth to which students engaged with these themes in their models.

The flow of information and programming representations were especially challenging for students, with fewer than 15% describing thinking around these topics [14]. Even though few students engaged with these topics in their model, they all did successfully build their own data displays using a working understanding of the flow of information and programming. The results from this study influenced our revisions and modifications made to the Sensor Immersion Unit to further highlight the importance of the flow of information and programming as well as how to use models to express ideas beyond typical scientific phenomena that students investigate in their science classrooms. These modifications included additional professional learning activities where teachers created incremental models of the DaSH and new teacher guides that provide support for teachers to implement and use student models to guide their instruction and student inquiry.

### 4.2. Debugging and Troubleshooting

As students assemble their DaSHs, they inevitably run into problems where it does not perform as they expect and they need to debug it to achieve the desired performance. Thus, as a fundamental part of using sensors and data-driven tools for scientific inquiry, students need to learn to troubleshoot and debug.

Teachers also need to learn to support students in troubleshooting and debugging the DaSH when they implement these units in their classes. This can be challenging for teachers as they are often new to the tools themselves.

To support students in learning these skills, the curricula provide wiring diagrams (Figure 6) that illustrate how to assemble the DaSH as well as programming tutorials (Figure 7). The MakeCode tutorials reduce the blocks students can access to a subset of all the blocks available and include color-coded hints to reduce the cognitive load required to figure out what block to use and where to find them. However, even with these scaffolds, students encounter bugs as they try to decide which blocks to use and program different actions with the DaSH. These bugs arise from issues with the hardware, software, and the bridge between them.

Our analysis of classroom experiences revealed common bugs faced by students, which we classified into three categories: hardware (HW), software (SW) and the intersection between the two (HW + SW) [10] (see Table 2). In teacher professional development sessions, we include targeted time where we highlight these common bugs and discuss strategies for helping students work through them.

Our analyses of classroom experiences also revealed the different ways teachers supported students during troubleshooting and debugging episodes. Analyses highlight three necessary practices for debugging and troubleshooting the DaSH: noticing there is an issue, finding the source of the issue, then fixing it. Therefore, helping students develop debugging skills also means helping them understand the connections between the DaSH’s operation, the hardware, and the software to diagnose when their expectation of the system (often connected to their programming logic) does not align with its output.

In interviews and professional development discussions, teachers talked about the different ways that students persisted (or not) when engaged in troubleshooting and debugging. For example, one teacher noted that her traditionally lower achieving class was much more successful than her traditionally higher achieving class in achieving the learning goals of the Sensor Immersion Unit because they were much better at working through mistakes and bugs. She shared that she needed to keep reminding her traditionally higher achieving class: “that’s kind of how you get to it to the end result… asking questions, making mistakes, troubleshooting, working with your group.” She also noted that her higher achieving students would become “nervous” when they did not know the right answer. Other teachers agreed that using the DaSH in their classrooms appropriately required ample time and motivation for students to tinker with and debug their systems to make them work.

### 4.3. Support for Place-Based Investigations

One purpose the DaSH serves is to support students to collect and analyze meaningful data from their environment, with the goal of getting them to ask the “so what” and “why here” questions. We have seen teachers uptake this goal in unique ways. For example, one teacher used the data display that anchored the Sensor Immersion Unit as evidence for getting more fans in her classroom during an early September heat wave (14). She described to one of the researchers during a classroom visit that she had shown her principal how the lights were blinking for the temperature reading (indicating that the temperature was over 30 degrees Celsius). Once she received the fans, she told her students how she had used the DaSH to make their room more comfortable, inspiring them to examine how to use the sound sensor to monitor the classroom noise level for their individual displays. 

During remote learning, due to the COVID-19 pandemic, several teachers distributed DaSHs to their students and conducted the Sensor Immersion Unit synchronously online. This allowed students to use the sensors to explore their homes and ask questions about why one student’s house was hot or cold or why there were higher than average carbon dioxide levels. In addition, one teacher encouraged her students to connect the battery packs to the micro:bit and explore their entire home and surrounding areas. Students created metal detectors using the magnetometer sensor on the micro:bit to search for items with strong magnetic fields. This led to quite an engaging conversation in the chat where students posted about the maximum possible magnetic reading they could find [16].

These examples illustrate how students are able to make personal connections with the data and use the DaSH to create displays that they can use to support their arguments. These are key aspects of being an informed consumer in this increasingly data-driven world.

## 5. Discussion and Conclusions

This article describes a sensor-based physical computing system (DaSH) using low-cost, portable, and accessible micro:bit/gator:bit microcontrollers. The system can be programmed to create information displays to introduce students to computing-infused scientific inquiry in ways that more closely resemble contemporary scientific practices. We also described two inquiry-oriented instructional units (Sensor Immersion and Composting Ecosystems) that guide students in learning to design the automated collection, processing and display of environmental data to address personally meaningful questions about the world around them. Finally, we described three major themes that emerged during analyses of participating teachers’ learning and classroom experiences.

As the sensor-based system and instructional units all cover content and pedagogy that is new to most middle school teachers, we have developed robust accompanying materials and activities for teacher professional development, called the CT-Integration cycle [23]. We have also conducted several empirical studies of student learning in the context of these units [14,17].

Implementing the physical computing software described above addresses an important need in students’ middle school science experiences. It offers an opportunity for integrated experiences in (1) computing, (2) engineering and (3) scientific processes by engaging students in the programming, wiring, and use of sensors that can help them interpret the world around them. Units were developed, and continue to be iterated, grounded in the fundamental concepts of project-based learning [27,28] that encourages longer, personally relevant, learning activities for students.

The DaSH and instructional units are intended to be adaptable to different middle school science teaching contexts. They are also opportunities for integrating computational thinking into science and technology classrooms. These experiences offer students opportunities to deeply engage with and understand the use and production of large streams of data [14]. In this way, this new educational technology creates opportunities for students to demystify sensor data, scientific data, and data streams in general by giving students the opportunity to engage in their production, analysis, and communication. By planning, debugging, and modeling their use of the DaSH, students learn to take ownership of the sensor data they collect and are able to collect data relevant to their lived experiences.

## Figures and Tables

**Figure 1 sensors-21-06243-f001:**
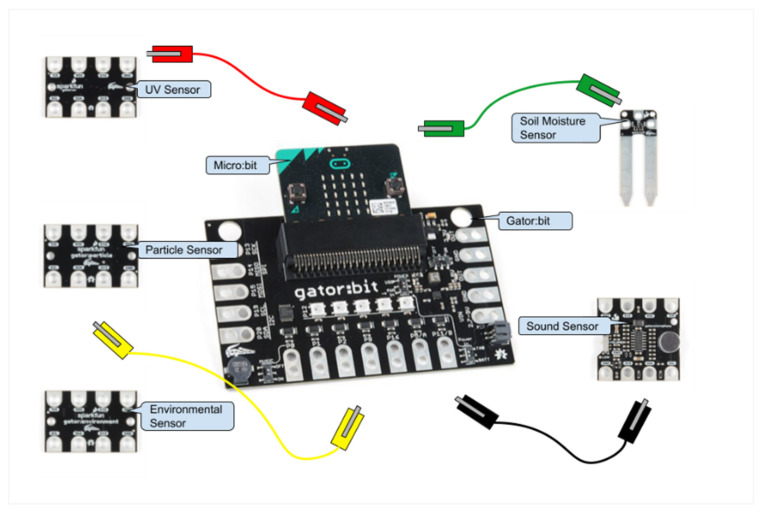
The components of the sensor-based physical computing system, the Data Sensor Hub (DaSH).

**Figure 2 sensors-21-06243-f002:**
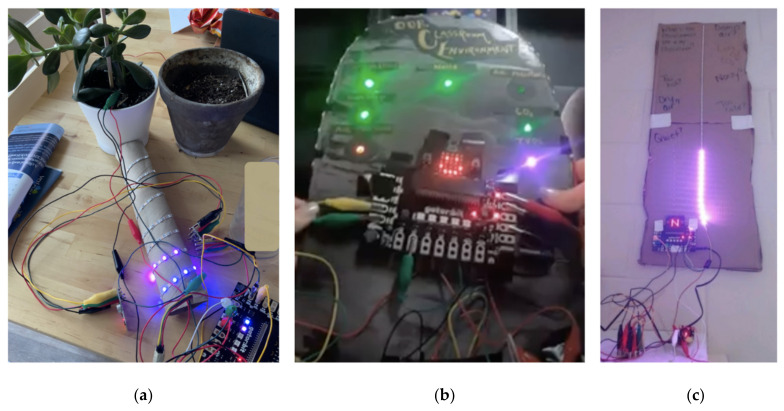
Three teacher-built DaSH examples: (**a**) monitoring a plant’s soil moisture and (**b**,**c**) monitoring the classroom environment (temperature, humidity, carbon dioxide, sound level).

**Figure 3 sensors-21-06243-f003:**
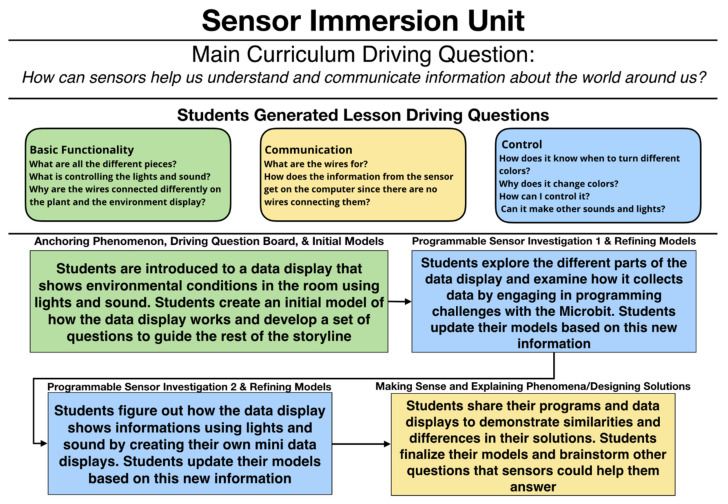
The overall flow of the Sensor Immersion Unit.

**Figure 4 sensors-21-06243-f004:**
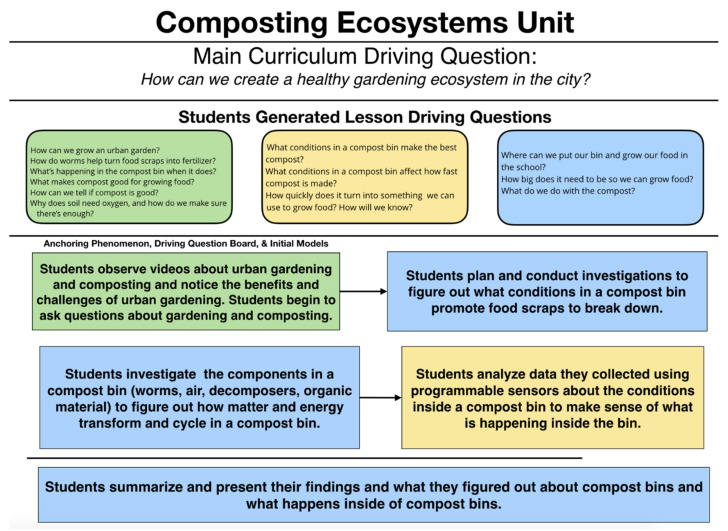
The overall flow of the Composting Ecosystems Unit and an overview of the content of each of the six lessons comprising the instructional unit.

**Figure 5 sensors-21-06243-f005:**
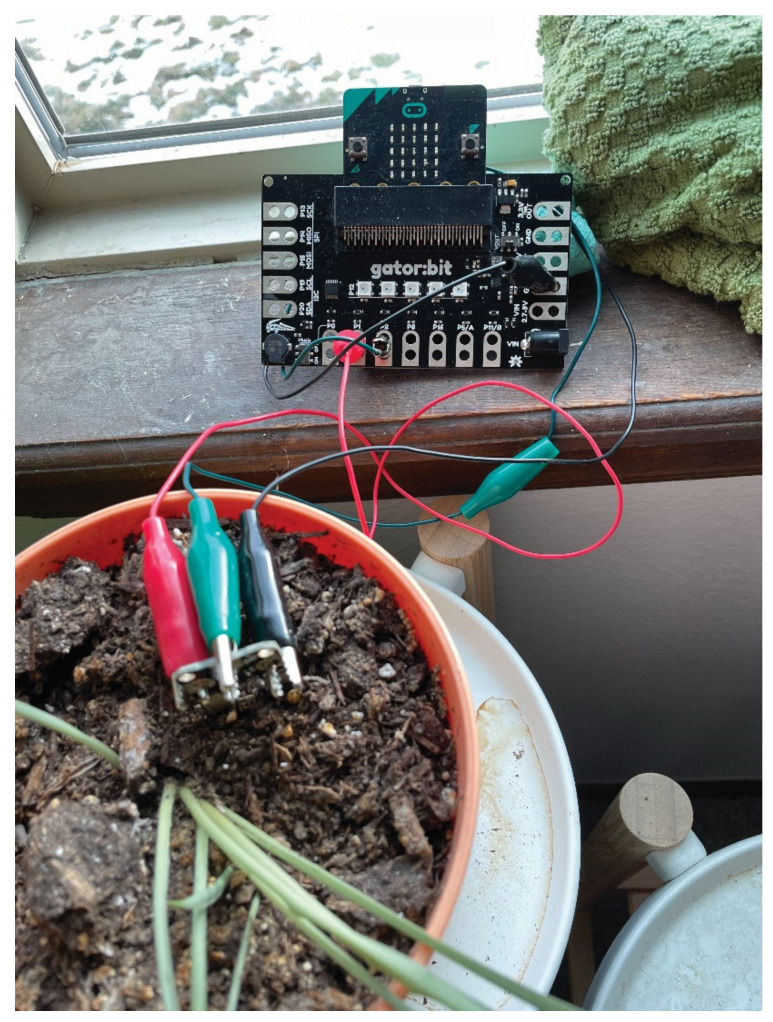
DaSH measuring soil moisture.

**Figure 6 sensors-21-06243-f006:**
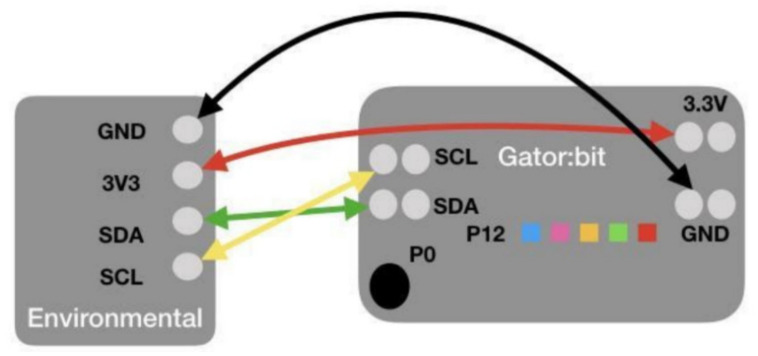
Wiring diagram for the environmental sensor.

**Figure 7 sensors-21-06243-f007:**
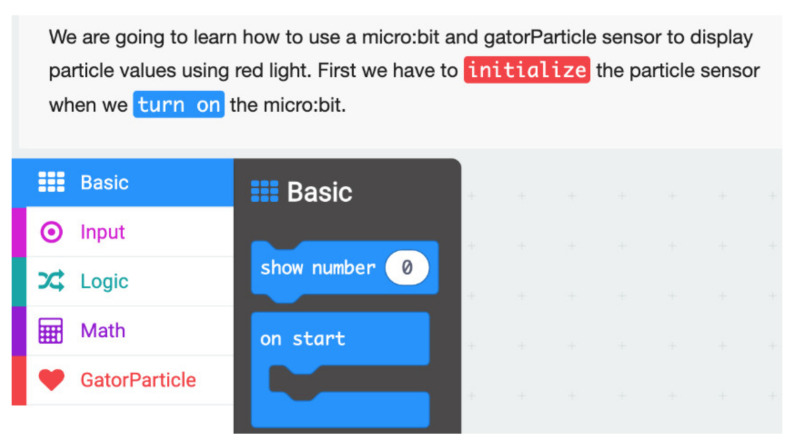
Screenshot of the first step in a tutorial for using the particle sensor. The instructions hint that the student will need to use a basic block and a gator particle block at this step. Only a subset of the blocks is visible to the student.

**Table 1 sensors-21-06243-t001:** Comparison between the cost of the DaSH and kits that can have similar functionality. Commercially available kits that included programmable sensors with educator friendly interfaces and learning materials were chosen as a comparison. Cost depends on what sensors are included and how students interact with the system. Prices retrieved on 3 September 2021 from https://www.thepocketlab.com/store, https://sphero.com/collections/all/family_littlebits+type_kit?sort_by=price-ascending, and Prices retrieved from https://www.vernier.com/product-category/?category=go-direct-packages&page_num=1.

Kit	DaSH	PocketLab	littleBits	Vernier Sensor Probes
**Cost**	EUR 90	EUR 33 to EUR 279	EUR 55 to EUR 335	EUR 144 to EUR 988

**Table 2 sensors-21-06243-t002:** Common errors in hardware, software, and combinations of hardware and software.

Bug	Description	Category
Assembly Errors	Students either forget to connect all the wires or connect the wires incorrectly.	HW
Power Issues	Insufficient power (from old batteries) leads to incorrect data readings and erroneous data. The gator:bit has a switch that controls the power out pins. If this switch is turned off, the sensor does not receive power and again leads to erroneous data.	HW
Transferring program from MakeCode to Micro:bit	Students have to download the file from MakeCode to their laptop and then transfer it to the micro:bit. Students are unfamiliar with this process and cannot easily tell if the process was completed correctly.	HW + SW
Code and wiring do not correspond	Some sensors require the connected pins to be specified in the program. The pins in the program must match the pins that the sensor is wired to.	HW + SW
Conditional Logic	Students include the most inclusive case in their if statement, making the code within the else if statements unreachable.	SW
Blocks not in scope	MakeCode is event driven so if students do include their code within an event handler, the code is shaded out and will not run.	SW
Sensor Values	Students first take a sensor reading and display it on the micro:bit screen. They then take another reading and use that reading for a conditional logic statement. The reading on the micro:bit does not always match the behavior of the logic statement because they use different sensor readings.	SW

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
