# Peer review of "The Data Sensor Hub (DaSH): A Physical Computing System to Support Middle School Inquiry Science Instruction"

_sensors, 2021, doi:10.3390/s21186243_

Round 1
Reviewer 1 Report
In summary, this work describes a sensor-based physical computing system, namely Data Sensor Hub (DaSH), which seemly has been used in some middle school science classes already. It basically leverages on the micro:bit and its peripherals, e.g., gator:bit, MakeCode, and additional sensors, to support teachers to design the lectures. Based on this, this work proposes two instructional units (Figure 7 and 8) to guide the designs of lectures.
However, the main concern of the reviewer is that, the position of this work is unclear. The paper more or less says: A has been done, B has been proven successful, so let's combine them. The paper proceeds explaining A and B in details, whose combination seems straightforward. Unfortunately, the reviewer cannot see the need for the proposed instructional units. What is the main issues of using DaSH? Is this work trying to resolve something concrete?
Considering the existing study in the literature, e.g., [16], [17], [22], the contribution of this work is limited. Moreover, the effectiveness of the proposed instructional units are not supported with sufficient evidences, neither qualitative nor quantitative. Suppose the focus of this work is on the proposed instructional units, Section 2 is definitely too long. More experimental or study cases are needed with the support of user feedback to prove the effectiveness of the concept.
Overall, the reviewer recommends rejecting the submission.
Detailed comments:
- The resolution of the figures is not sufficient for publications, e.g., Figure 1b, 2, 9, 10.
- Hardy and colleagues -> Hardy et al.
- Why does Section 2.5 explain again Figure 1?
Reviewer 2 Report
- The work presents a sensor-based physical computing system, called the Data Sensor Hub (DaSH), which permits to process data from different sensors and allows students to immerse in the application of these sensors for monitoring different processes and systems.
- The document is well written and the information is neatly structured.
- One of the claims made by the authors about the presented device is that "as a technology for public schools, it must be affordable. Currently, a micro:bit sells for approximately €12." However, later on the document, the authors include the gator:bit as part of the proposed platform which costs around €18. For that regard, it should be included a brief section with price comparisons between the proposed platform and other similar options available.
- Figure 10 has low resolution and is really hard to understand its content.
- Pedagogically, it is clear the basis for the instructional design of the proposal. Nonetheless, it could be helpful to include some experimental results with student groups to validate the effectiveness of the proposed system.
Reviewer 3 Report
The topic is interesting, and the authors introduce that motivates the reading of the study.
A review of recent literature is done, however, you must comment more on the results and gaps of those studies, and this research will cover that.
Add a research question or objective in the introduction.
As a proposal to support classes, it is interesting and they explain in detail the educational intervention, which facilitates its replication in other contexts. However, methodologically, it is necessary to define the type of study involved; it seems more qualitative and observational. Here they should describe the instrument they used to record the data: rubric for example. They should also present results in the form of categories of analysis, for example, to demonstrate that these benefits did occur for the students.
Once the methodological part and the results have been corrected, the discussion and conclusions section should be revised based on these evidences.

Round 2
Reviewer 1 Report
Thanks. The authors have addressed basically all my concerns, especially on the position of this work. I think adding Section 4 to split the case study is a good idea.
One minor comment: I see that Section 2.1.1- 2.1.3 are about different attempts, but the intention is not clear for readers, e.g., what is iteration? After listing criteria in Section 2.1, I guess a brief locally connection is needed. Similarly, Section 3 before Section 3.1 might also need a short opening paragraph to brief the structure of this section.
Author Response
- "I see that Section 2.1.1- 2.1.3 are about different attempts, but the intention is not clear for readers, e.g., what is iteration? After listing criteria in Section 2.1, I guess a brief locally connection is needed."
We changed the language to more clearly reflect the design process. We are now referring to the versions of the DaSH that were developed during iterative design cycles. We also added a paragraph describing the content of Section 2.1.
- Similarly, Section 3 before Section 3.1 might also need a short opening paragraph to brief the structure of this section
Thank you for this suggestion. We have added a brief paragraph introducing the purposes of Section 3, which is depicting two sample Units that we have co-designed with teachers to use the DaSH systems in their classrooms. We believe this addition makes the structure of the section more clear to the reader.
Reviewer 3 Report
I think that the authors made the suggested modifications. Thank you for your attention.
Author Response
Thank you for reading our revised manuscript. And thank you for very clear feedback of the first submission.